# Recurrent Metastatic Pulmonary Synovial Sarcoma during Pregnancy: A Case Report and Literature Review

**DOI:** 10.3390/diagnostics14040424

**Published:** 2024-02-14

**Authors:** Silvia De Rocco, Jasmine Di Biasi, Ilaria Fantasia, Sara Tabacco, Enrico Ricevuto, Pierpaolo Palumbo, Ilenia Imbergamo, Manuela Ludovisi, Maurizio Guido

**Affiliations:** 1Unit of Obstetrics and Gynecology, San Salvatore Hospital, 67100 L’Aquila, Italy; silviaderocco2@gmail.com (S.D.R.); jasmine.dibiasi@aol.com (J.D.B.); ilariafantasia@gmail.com (I.F.); saratabacco87@gmail.com (S.T.); ilenia.imbergamo@virgilio.it (I.I.); maurizio.guido@univaq.it (M.G.); 2Department of Life, Health and Environmental Sciences, University of L’Aquila, 67100 L’Aquila, Italy; 3Oncology Territorial Care, San Salvatore Hospital, Oncology Network ASL1 Abruzzo, University of L’Aquila, 67100 L’Aquila, Italy; enrico.ricevuto@univaq.it; 4Department of Biotechnological and Applied Clinical Sciences, University of L’Aquila, 67100 L’Aquila, Italy; palumbopierpaolo89@gmail.com

**Keywords:** synovial sarcoma, pregnancy, chemotherapy, radiological imaging

## Abstract

Primary pulmonary synovial sarcoma is a rare type of soft tissue tumor. Exceptionally it can occur during pregnancy, representing a challenge in management and treatment given its notable aggressiveness and the not infrequent incidence of maternal death. We report our case of metastatic recurrence of pulmonary synovial sarcoma during pregnancy, with the aim to emphasize the decision-making, diagnostic, and therapeutic multidisciplinary processes and the evolution of the pathology. Besides, we focused on the analysis of the limited literature data available on the topic.

## 1. Introduction

Soft tissue sarcoma is a rare type of cancer affecting about 1% of cancer patients with almost double the incidence in male patients and an estimated 900 new cases in women in 2020 [1] Primary pulmonary synovial sarcoma (PPSS) constitutes a very uncommon and highly aggressive histotype, accounting for approximately 8% of soft tissue sarcomas and <0.5% of lung tumors [2,3] Worldwide only 4 cases of PPSS have been reported during pregnancy [4,5,6,7] We intend to present the case of a terminally ill woman with metastatic recurrence during pregnancy. We also carried out a thorough review of the literature. 

## 2. Case Presentation

A 43-year-old patient, in her fourth spontaneous pregnancy, was referred to our Unit at 24 weeks’ gestation with a history of recurrent metastatic synovial sarcoma. In 2009 the patient was diagnosed with monophasic synovial sarcoma after right pulmonary lobectomy and ilo-mediastinal lymphadenectomy. Surgery was followed by 5 cycles of adjuvant chemotherapy based on Epirubicine, Isofosfamide, and Mercaptoethansulfonate. In 2014 she had her first renal recurrence treated with left nephrectomy, followed by a second recurrence in 2021 treated with partial resection of right scapula and radiotherapy. Postoperative checks at CT scan showed a good response to therapy of the disease (Figure 1).

During subsequent checks in 2022, multiple right lung metastases and homolateral lymph node localization were found, requesting surgical excision of lymph nodal supraclavicular mass. A few weeks after the surgery, the patient found out she was pregnant. Despite she was fully informed by the oncologists that chemotherapy was not contraindicated in pregnancy, she decided to carry on with the pregnancy and to refuse the therapy. By the time we took charge of the patient at 24 weeks, the pregnancy was uneventful, and the patient didn’t report any specific symptoms. The first trimester screening for fetal aneuploidies was low risk and the anomaly scan was normal with normal fetal growth and Dopplers. After multidisciplinary counseling, the patient confirmed her intention not to pursue treatments. Thromboprophylaxis with 4000 UI of heparin was started. Follow-up appointments were arranged every 2 weeks by the High-Risk Pregnancy clinic along with her oncology follow-up. At 28 weeks the patient began to report pain in the scapular site of the metastasis, which was treated with paracetamol and 5 mg per day of methylprednisolone with partial improvement of symptoms. In agreement with the oncologist, a CT scan was scheduled at 32 weeks which showed an increase in the size of pulmonary metastases. At 34 weeks gestation, the patient began to develop dyspnea along with worsening pain for which she was hospitalized. Her vitals were normal, and no oxygen support was needed. Fetal conditions were stable. Considering the worsening of the maternal clinical condition and the gestational age reached, a full course of steroid prophylaxis with betamethasone was done and a cesarean section was performed thereafter. The newborn weight was 2070 gr and the clinical conditions were normal. The postpartum course was uneventful, and the patient was discharged five days after delivery. Seventeen days after the cesarean section the patient was hospitalized for severe respiratory failure. The vital signs showed an oxygen saturation level of 95% on 4L/min O_2_ flow, heart rate of 115 beats per minute, reduced vesicular murmur over the entire pulmonary area, and absence at baseline level bilaterally. Chest CT scans showed pleural carcinosis and extensive metastatic tissue component occupying the entire lower right lung lobe, an increase in size and number of the remaining pulmonary nodules with compressive phenomena on the trachea and esophagus, and osteolysis of the right scapula (Figure 2 and Figure 3). Chemotherapy with Gemcitabine and Docetaxel was started but after two cycles another hospitalization for respiratory failure occurred, as a result of which the woman died.

## 3. Discussion

Synovial sarcoma is a very rare and aggressive type of soft tissue cancer (STC) accounting for 5% to 10% of all STS [8]. It is a mesenchymal spindle cell tumor that displays variable epithelial differentiation, including glandular formation, and has a specific chromosomal translocation t(X;18) (p11;q11) resulting in the SYT-SSX fusion protein in more than 95% of cases [9,10]. Synovial sarcoma does not arise from synovial tissue; nevertheless, the histological figures mimic the appearance of a developing synovium [11]. Three subtypes of synovial sarcoma are described (monophasic, biphasic, and poorly differentiated) based on the type and percentage of cell population present. The monophasic one has only the spindle cell component [10]. In the vast majority of cases, it affects the extremities, especially the knee and it is prevalent in adolescents and young adults between 10 and 40 years of age [3]. Although the lung is a common site of metastasis in the clinical course of the disease, the primary location of the disease has only recently been recognized as a primary tumor [12,13]. Therefore, diagnosis of PPSS is based on clinical, radiological (the gold standard is magnetic resonance imaging [14], pathological, and immunohistochemical examinations to exclude other primary tumors and metastatic sarcomas [15]. Typical presenting symptoms include dyspnea (8–36%), chest pain (24–80%), cough (10–33%), and hemoptysis (20–25%). Less commonly reported symptoms include shoulder or back pain, fever, and extremity swelling [4,16]. The standard treatment is surgical eradication. Neoadjuvant radiotherapy and/or chemotherapy are reserved for locally advanced tumor-invading critical structures [17]. Up to 50% of synovial sarcoma recur, usually within 2 years, but sometimes up to 30 years after diagnosis. The five-year survival rate is 36–76%, and the 10-year survival rate is 20–63% [3]. Krieg et al. demonstrated that local recurrence in synovial sarcoma occurs after a mean of 3.6 years (range 0.5–15 years) while metastasis occurred at a mean of 5.7 years (range 0.5–16.3 years) [18]. Prognostic factors are disease stage, tumor size, and tumor grade. The best outcomes are in childhood patients, in tumors that are <5 cm in diameter, have <10 mitoses/10 hpf and no necrosis, and when the tumor is eradicated locally [19,20]. To date, fewer than 100 cases have been reported in the literature and only four primary synovial sarcomas during pregnancy have been encountered and are summarized in Table 1 [4,7].

Sakurai et al. described the case of a 33-year-old woman presenting with acute back pain, left hemothorax and a lung mass in her fifth month of pregnancy. She underwent surgical evacuation of the hemothorax and resection of the tumor. The pregnancy resulted in miscarriage and she declined further treatments [7]. Esaka et al. in 2008 reported another case of pulmonary synovial sarcoma diagnosed during pregnancy and presented with anemia and progressive dyspnea due to a pneumothorax in the 34th week of pregnancy. Delivery was induced, allowing for adjuvant chemotherapy. However, the patient died after 13 months of diagnosis [4]. Bunch et al. reported the case of a 38-year-old woman at 26 weeks of gestation with the finding of a lung mass. Right pneumonectomy was performed after two weeks due to worsening dyspnea and a cesarean section was performed at 32 weeks for fetal growth restriction. Puerperium was complicated by pulmonary embolism and tumor growth. She underwent radiotherapy, but despite intensive care, she died 6 weeks after delivery [6]. Finally, Harris et al. described the case of a 26-year-old female with primary pulmonary synovial sarcoma treated with chemotherapy during pregnancy. The diagnosis was made following investigations performed for the appearance of cough and pleuritic chest pain. Neoadjuvant chemotherapy with Ifosfamide was focused on containing the growth of the mass. At 31 weeks, an emergency cesarean section was performed for non-reassuring fetal tracing. Two more cycles of chemotherapy were practiced, then she was then scheduled for right total pneumonectomy and mediastinal lymphadenectomy approximately 8 weeks after delivery. She died about 13 months after the diagnosis [5]. The peculiarity of this case is the very long survival time from the first diagnosis compared with data reported in the literature. In fact, from the first diagnosis, the woman had a survival time of 13 years. In addition, the survival time was characterized by multiple recurrences that were treated not only with chemotherapy regimens but also with surgery to remove the neoplastic masses and by subsequent radiotherapy treatments. Complicating the clinical picture was the patient’s lack of compliance with the treatment choices made by the healthcare providers. The patient, in agreement with her husband, refused chemotherapy treatment during pregnancy and then resumed it only after delivery. However, the analysis of data present in the literature, although based on a few case reports, shows that the pregnancy’s outcome tends to be similar in primary and recurrent forms. In particular, the patient must be informed of the increased risk of preterm birth, either on maternal or fetal indications, in the third trimester of pregnancy.

Multidisciplinary counseling has a fundamental role in these cases. The diagnosis of cancer during pregnancy can result in an overlap of feelings of despair and fear of death with feelings of joy and the normal stress of becoming a mother. Recognizing the early signs of distress in these women is essential to refer them early to the necessary psychosocial services. This aspect is even more important when considering the possible effects of stress during pregnancy, such as preterm birth, low birth weight, and neurodevelopmental disorders and disabilities [21]. Women may be particularly at risk of experiencing high levels of long-term distress when they are diagnosed with cancer during pregnancy. That is why the physician becomes fundamental in paying attention to these initial aspects of distress as indicative of the need to seek psychological support [22]. Crucial in the pathway was the role played as a team, for the management of pregnancy and pathology. Palliative therapies for pain management together with the oncologist and the pain therapist, made it possible in this case to reach an adequate gestational age for the birth of the child when the maternal clinical conditions had worsened. 

## 4. Conclusions

To our knowledge, this is the first case of multi-recurring and metastatic form of primary synovial sarcoma of the lung recorded in the literature. Primary lung involvement is usually rare in synovial sarcoma and is burdened with a poor prognosis. When diagnosed during pregnancy, the patient must be managed with a multidisciplinary approach to identify the most appropriate treatments, curative or palliative, respecting her choices. The risk of adverse pregnancy and maternal outcomes must be discussed, and in particular the risk of preterm birth for maternal or fetal indication.

## Figures and Tables

**Figure 1 diagnostics-14-00424-f001:**
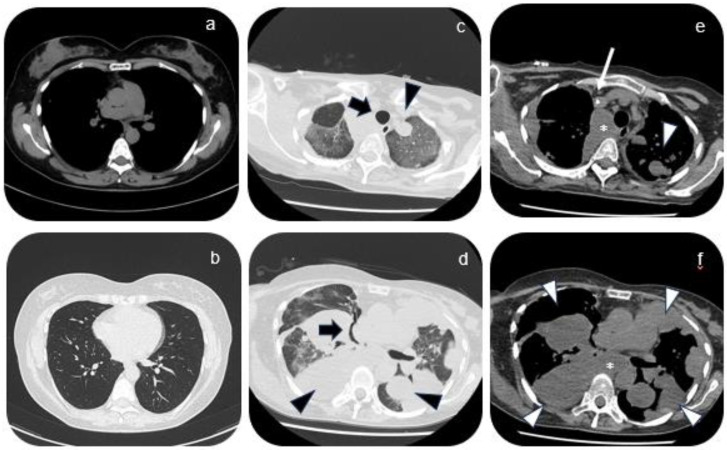
Images (**a**) and (**b**) show axial views of non-contrast enhanced lung CT acquired during the follow-up before the pregnancy. In image a, mediastinal window. In images (**b**) lung window. CT images show a regular representation of the lung parenchyma. In the mediastinal window, no suspicious lymphadenomegalies are highlighted. Images from (**c**) to (**f**) show axial views of non-enhanced lung CT acquired after the pregnancy. In images (**c**) and (**d**), lung window. In image (**e**) and (**f**), mediastinal window. CT scan after the pregnancy shows multiple masses within the lung parenchyma (black and white arrowheads). Widespread areas of ground glass-like lung density can be seen in both lungs, resulting from possible interstitial congestion. In this regard, diffuse mediastinal lymphadenopathy can be seen in the mediastinal window (white asterisks). Compressive phenomena affecting the superior right bronchus could be appreciated in image (**d**); trachea appears slightly dislocated on the left (thick black arrows). White thin arrow points the presence of a central catheter (image (**e**)).

**Figure 2 diagnostics-14-00424-f002:**
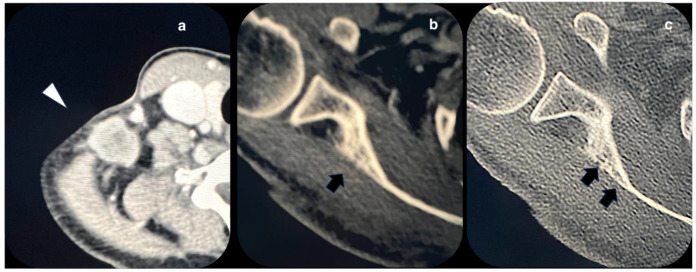
Axial views of contrast-enhanced CT acquired after pregnancy. Images (**a**), (**b**), and (**c**) show views of the right shoulder and homolateral supraclavicular region. Images show enlarged lymphadenopathy with uneven vascularization due to the presence of areas of central colliquation (white arrowhead). Images (**b**) and (**c**) show a detail of the right scapula. In image (**c**), a moth-eaten osteolysis interrupts the posterior cortical surface (black arrows), due to a bone metastasis.

**Figure 3 diagnostics-14-00424-f003:**
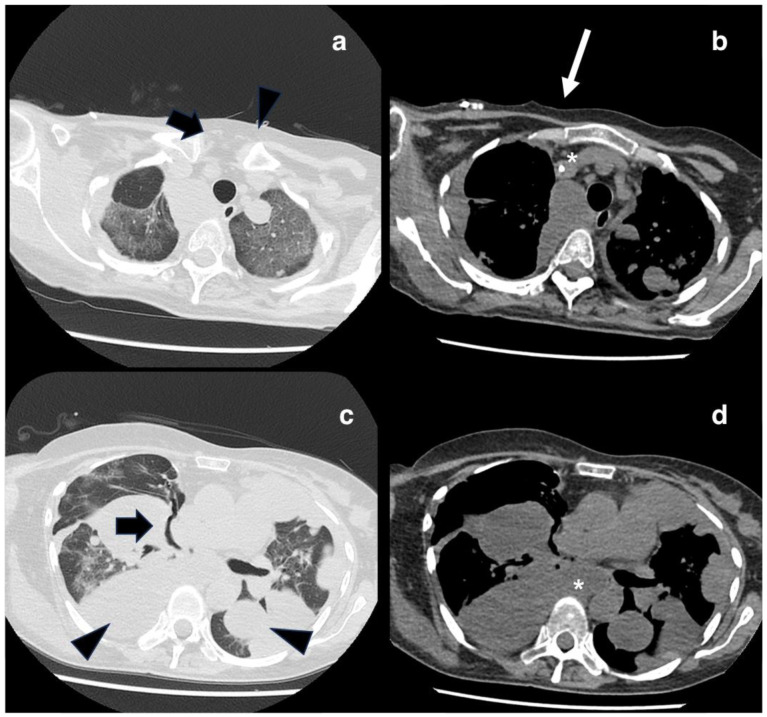
Axial views of non-enhanced lung CT acquired after the pregnancy. In images (**a**) and (**c**), lung window. In images (**b**) and (**d**), the mediastinal window. The CT scan shows multiple nodular lesions within the lung parenchyma (black arrowhead). Lymphadenopathy can be seen in the mediastinal window (white asterisks). Compressive phenomena affecting the superior right bronchus could be appreciated in image (**c**); the trachea appears slightly dislocated on the left (thick black arrow). A white thin arrow points to the presence of a central catheter (image (**b**)).

**Table 1 diagnostics-14-00424-t001:** Articles reporting on the occurrence of synovial sarcoma during pregnancy.

Authors	Primary	Diagnosis	Treatments	Pregnancy Outcome	GA at Delivery	Mode of Delivery	Maternal Outcome after Diagnosis	Maternal Outcome after Delivery
Sakurai H et al., 2006 [7]	Yes	During pregnancy	Refused	Miscarriage	20 weeks	/	Unknown	Unknown
Esaka EJ et al., 2008 [4]	Yes	During pregnancy	Performed	Live birth	34 weeks	Spontaneous	Death after 13 months of diagnosis	Death after 13 months of delivery
Bunch K et al., 2012 [6]	Yes	During pregnancy	Performed	Live birth	32 weeks	CS for fetal conditions	Death after 12 weeks of diagnosis	Death after 6 weeks of delivery
Harris EM et al., 2014 [5]	Yes	During pregnancy	Performed	Live birth	31 weeks	CS for fetal conditions	Death after 12 weeks of diagnosis	Death after 6 weeks of delivery
Present case	Recurrence	Before pregnancy	Refused	Live birth	34 weeks	CS for maternal conditions	Death 13 years after diagnosis	Death after 3 months of delivery

GA, gestational age; CS, cesarean section.

## Data Availability

Not applicable.

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
