# Peer review of "Recurrent Metastatic Pulmonary Synovial Sarcoma during Pregnancy: A Case Report and Literature Review"

_diagnostics, 2024, doi:10.3390/diagnostics14040424_

Round 1

Reviewer 1 Report

Comments and Suggestions for Authors

The authors report a clinical case dealing with a rare type of pulmonary tumor and its evolution during pregnancy – a topic potentially interesting to both oncology and obstetrics audiences. The case is reasonably clearly presented and the discussion section seems scientifically sound.

Here are some suggestions on how to improve the manuscript

Title seems misleading – as the recurrence of MPSS seems to precede the pregnancy

Please rephrase: Line 30 ‘along with a literature review of the literature.’

Line 39 ‘Postoperative checks at CT scan showed a good regression of the disease (Figure 1)’ is somewhat unclear; the term regression seems inadequate.

Figure 1. – is somewhat redundant as it shows normal findings; might be useful if shown in conjunction with figure 3 making sure the images are leveled.

Figure 2. legend – ‘sovraclavear region’ should probably be changed to supraclavicular region

Figure 3 – legend – nodule (pulmonary) is used when the lesion diameter is less than 3 cm – ‘mass’ seems to be the adequate term

-          The asterisk in image b seems to be placed on a vascular structure

-          The thick black arrow does not point to trachea either in image a or c

-          Mediastinal adenopathies are usually reported by indicating group – 2R (probably) in image a, 7 in image d

-          Image a shows some interstitial anomalies – perhaps they explain the respiratory failure

Some general suggestions:

Would be potentially useful to list available therapeutic strategies and discuss potential impact during pregnancy; there are some novel agents such as TKIs showing potential.

Would be interesting to explore potential mechanisms linking pregnancy to SS18-SSX expression or SWI/SNF complex (if such data is available).

Comments on the Quality of English Language

.

Author Response

1. Title seems misleading – as the recurrence of MPSS seems to precede the pregnancy. We believe that the title is not misleading as the patient suffered a different number of relapses both outside and during pregnancy. Can we keep this title or do you prefer the title: "Multiple Recurrent metastatic pulmonary synovial sarcoma during and outside pregnancy: a case report and literature review"?

2. Please rephrase: Line 30 ‘along with a literature review of the literature.’. We rephraded Line 30: We intend to present the case of a terminally ill woman with metastatic recurrence during pregnancy. We also carried out a thorough review of the literature.
3. Line 39 ‘Postoperative checks at CT scan showed a good regression of the disease (Figure 1)’ is somewhat unclear; the term regression seems inadequate.
We have replaced the term regression with the term "good response to therapy".
4. Figure 2. legend – ‘sovraclavear region’ should probably be changed to supraclavicular region. We have replaced the term ‘sovraclavear region’ with the term 'supraclavicular region'.
5. Figure 3 – legend – nodule (pulmonary) is used when the lesion diameter is less than 3 cm – ‘mass’ seems to be the adequate term. We have replaced the term ‘multiple nodular lesion’ with the term 'masses'.
6. W
e corrected the formatting of the images and made the required changes

Please see the attachment for required changes

Reviewer 2 Report

Comments and Suggestions for Authors

 This case report describes pulmonary synovial sarcoma during pregnancy. This was a very rare case report, and the course was summarized in an easy-to-understand manner. There did not appear to be any areas that required major modifications. Please consider some minor modifications.

1. If you can confirm the CT value (WW, WL) in figure 1-3, please include it.

2. Please confirm if are the arrows and asterisks in Figure 3 misaligned.

3. I thought it would be easier to understand if the maternal outcomes in Table 1 were unified as ‘death after delivery’. If you would like to include ‘death after diagnosis’, please consider adding a new column.

Author Response

1. We corrected the formatting of the images and made the required changes

2. I thought it would be easier to understand if the maternal outcomes in Table 1 were unified as ‘death after delivery’. If you would like to include ‘death after diagnosis’, please consider adding a new column. We divided maternal outcomes into outcomes after delivery and after diagnosis

Please see the attachment for required changes
